https://doi.org/10.1038/s41467-020-16508-x · **OPEN**

# Electrode interface optimization advances conversion efficiency and stability of thermoelectric devices

Jing Chu[1,2,3], Jian Huang[1,3], Ruiheng Liu [1,2✉], Jincheng Liao[1], Xugui Xia[1], Qihao Zhang[1], Chao Wang[1], Ming Gu[1], Shengqiang Bai [1,2✉], Xun Shi [1] & Lidong Chen [1,2]

Although the $CoSb_3$-based skutterudite thermoelectric devices have been highly expected for wide uses such as waste heat recovery and space power supply, the limited long-term service stability majorly determined by the degradation of electrode interface obstructs its applications. Here, we built up an effective criterion for screening barrier layer based on the combination of negative interfacial reaction energy and high activation energy barrier of Sb migration through the formed interfacial reaction layer. Accordingly, we predicted niobium as a promising barrier layer. The experimental results show the skutterudite/Nb joint has the slowest interfacial reaction layer growth rate and smallest interfacial electrical resistivity. The fabricated 8-pair skutterudite module using Nb as barrier layer achieves a recorded conversion efficiency of 10.2% at hot-side temperature of 872 K and shows excellent stability during long-time aging. This simple criterion provides an effective guidance on screening barrier layer with bonding-blocking-conducting synergetic functions for thermoelectric device integration.

[1] State Key Laboratory of High Performance Ceramics and Superfine Microstructure, Shanghai Institute of Ceramics, Chinese Academy of Sciences, Shanghai 200050, China. [2] Center of Materials Science and Optoelectronics Engineering, University of Chinese Academy of Sciences, Beijing 100049, China. [3] These authors contributed equally: Jing Chu, Jian Huang. ✉email: liurh@mail.sic.ac.cn; bsq@mail.sic.ac.cn

D ue to the long-term reliability, thermoelectric (TE) generators have been successfully employed in various planetary explorations and deep space missions since 1960s (refs. [1–3]). For example, Voyager 1 and Voyager 2, powered by radioisotope thermoelectric generators (RTG) using SiGe-based alloys for over 40 years, have successively flown out of the solar system. In recent years, because of the world-wide demand for industrial waste heat recovery in the fields, such as steelworks, cement factories, and automobiles powered by internal combustion engines[4–7], TE technology has attracted ever-rising attentions from both academic and industrial communities[8]. In the past two decades, the dimensionless figure of merit ($ZT$) of TE materials[8–13], such as filled skutterudites (SKDs), nano-PbTe, half-Heusler compounds, liquid-like materials, and Zintl phase, has been greatly promoted over 1.5 and sporadically over 2.0. Significant achievements have also been gained on the design and integration of high-performance devices. The record of device conversion efficiency has been successively renewed up to 9% for single-stage modules, and even across 12% for segmented and cascaded modules[14–18]. These achievements strengthen the promising potential for widespread applications of TE devices in waste heat recovery, and also promote next generation of RTGs for long-term space exploration.

When TE materials are fabricated into devices, besides the TE property of materials and the topologic structure design, electrode interface[19] plays a vital role in the output performance and long-term reliability. The influence of the interfacial resistivity on conversion efficiency, without considering interfacial thermal resistivity, can be given[20] by $\eta = \frac{\left(\frac{T_h}{T_h - T_c}\right)}{2 - \frac{1}{2}\left(\frac{T_h}{T_h - T_c}\right) + \frac{4}{ZT_h}\left(1 + \frac{\lambda}{L}\right)}$, where $T_h$ and $T_c$ are the temperatures of hot side and cold side, respectively, $\lambda$ is the interfacial resistivity parameter ($\lambda = \frac{4R}{[R]}$, where $R$ is the interfacial resistivity and $[R]$ is the electrical resistivity of TE materials), and $L$ is the length of TE legs. Therefore, to tap the potential of the constituent TE materials, the interfacial resistivity is expected to be as low as possible. Besides, during long-term service, element diffusion, and reactions at electrode interface usually produce interfacial reaction layers (IRLs), leading to additional electrical and thermal resistance[21], and residual stress[22,23], thus greatly deteriorate the performance of TE devices.

The general solution is to introduce a metallic barrier layer between the electrode and TE material that can suppress elements diffusion and reactions[24–26]. Traditionally, the common-sense principles, including thermal expansion match for mechanical reliability and work function match for low initial contact resistance[27], have been used for selecting candidates of barrier layers. So far, a kind of TE modules have been developed using various metals or alloys as barrier layer selected by such common-sense principles qualitatively. However, this trial-and-error method is not only born to be time-and-cost-consuming, but also hard to achieve optimized bonding. Currently, there is a lack of effective method to predict interfacial structure evolution in long-term service, and the dynamic approaches to modeling interfacial behavior is far from guiding the optimization of TE devices. The interfacial resistance and stability have become the stumbling block to the large-scale applications of TE materials and devices.

Among the state-of-the-art TE materials, $CoSb_3$-based filled SKDs are the most promising TE materials for the power generation in middle temperature range. However, the long-term stability and its predictability of the SKD device need further improvement. Especially, theoretical model to screen and discover interface materials are urgently required because all the current approaches are just based on the trial-and-error method.

Here, we report on a systematic investigation on the interface behavior by taking into account the interfacial reactions and element diffusion. Combining the first-principles calculations and experimental results, we propose a criterion, the combination of interfacial reaction energy ($E_{IR}$) and activation energy barrier of Sb migration ($E_{Mig}$) in the formed IRL, to determine the interfacial reliability of SKD/metal bonding. And then, a sweet spot to predict promising barrier layer candidates is obtained. According to this screening criterion, we predict niobium (Nb) as one of promising candidates of barrier layer, which has a negative $E_{IR}$ and a large $E_{Mig}$ for effectively blocking. By building the bridge between microscopic kinetic factors and macroscopic features of the IRL, we successfully figure out the interfacial resistivities at various service durations and temperatures for various metal/SKD joints. Experimentally, SKD/Nb joint (Nb joint) shows the slowest IRL growth rate and smallest interfacial resistivity among all reported SKD/metal joints[28]. Finally, combined with the topologic structure design, an eight-pair SKD module is fabricated adopting Nb as the barrier layer and a record high conversion efficiency of 10.2% in the single-stage module is achieved at hot-side temperature of 872 K. Meanwhile, the extremely low diffusion rate also ensures the excellent long-term stability of the module, which is verified by a negligible degradation (<1%) of both power output and conversion efficiency after 846 h service test at high temperature.

## Results

**Screening criterion for barrier layers**. According to the theory of diffusion reaction[29] (Fig. 1a, taking Nb joint as the example), the growth mechanism of the IRL involves two consecutive steps, the chemical reaction step and the diffusion step. In the initial stage of the chemical reaction, the constituent atoms from both sides may easily come into contact with each other, and then the enthalpy change of bond reconstruction results in the formation of new stable phases. Since no Yb is experimentally observed in the growth of IRL (ref. [30]), and the low filling fraction and the small diffusion coefficient of Yb in $CoSb_3$, it is reasonable to simplify Yb-filled $CoSb_3$-based SKD to $CoSb_3$ in theoretical model for the analysis of interfacial microstructure evolution, although a few filler atoms, such as Ba are reported to participate the IRL growth[31]. Considering the essential requirements as a barrier layer of having enough high melting point and being inertia as dopant in $CoSb_3$-based-filled SKD, a series of transition metals with $d$ electrons (Sc, Y, Ti, zirconium (Zr), Hf, Nb, Ta, Mo, W, Cu, Au, all metals are referred to as X) are regarded as the possible barrier layer candidates. Also, because Sb has the highest diffusion coefficient among all elements, Sb will move from the SKDs to transition metals for such SKD/transition metal interface. Therefore, at the initial stage of IRL formation, the interfacial chemical reactions between SKD and transition metals are $CoSb_3 + X \rightarrow CoSb_2 + XSb_y$. And the formation energy of interfacial chemical reactions, defined as the interfacial reaction energy, $E_{IR}$, is $E_{IR} = E_{CoSb2} + E_{XSby} - E_{CoSb3} - E_X$. Such reaction is dynamically determined by Sb diffusion. Thus, it can be considered as the case of Sb rich, especially when the thickness of barrier layer material is quite small as compared with the vast thickness of SKDs. For element Nb, the $y$ is 2 and the final product phase is $NbSb_2$. The schematic model for such reaction is shown in Fig. 1a. Definitely, in order to obtain a good contact between SKD and X, $E_{IR}$ should be negative. Otherwise, no chemical reactions and bonds are formed, leading to poor mechanical strength for the interface. Based on the calculations, we found that Cu and W can't react with SKDs, and they are not potential barrier layer candidates. Instead, many other transition

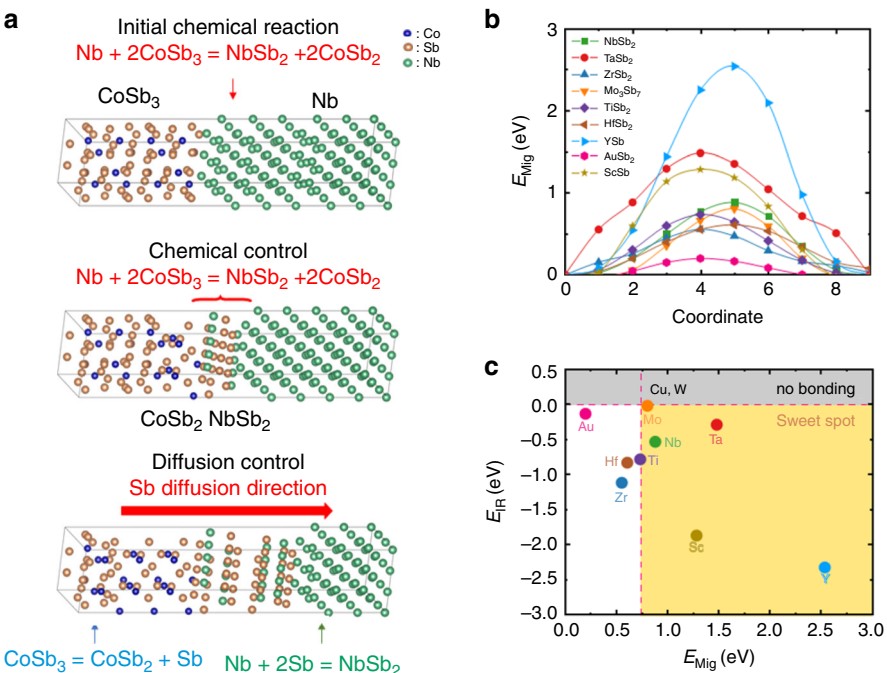

**Fig. 1 Screening criterion for barrier layers. a** The schematic model of IRL growth, taking Nb joint as the example. **b** The activation energy barrier of Sb migration ($E_{Mig}$) to neighboring Sb vacancy. **c** The correlated map of interfacial reaction energy ($E_{IR}$) and activation energy barrier ($E_{Mig}$). Values of $E_{IR}$ and $E_{Mig}$ are shown in Supplementary Table 1. The elements in the area (called sweet spot) with negative $E_{IR}$ and larger $E_{Mig}$ (similar to or larger than Ti) are believed potential barrier layer candidates.

metals, such as Nb, Ti, and Zr have negative $E_{IR}$, which can be used as possible barrier layer materials.

As the adjected reactant atoms are exhausted with the interfacial reaction proceeding, the atomic diffusion shall become dominant for the growth of the IRL (Fig. 1a). In SKDs, the interface materials include $CoSb_2$ and $XSb_y$ compounds. In the previous studies[21,30], we found that element Sb has the largest diffusion coefficient. In addition, $CoSb_2$ is a metallic compound with very good electrical conductivity[32] and reasonable large Sb diffusion coefficient. Thus, in this study, only the diffusion of Sb in the formed $XSb_y$ compounds is considered.

We thus believe that the activation energy barrier of Sb migration, $E_{Mig}$, in $XSb_y$ compounds principally determines the growth rate of the IRLs. The lower value of $E_{Mig}$ means the faster migration (diffusion) of atoms, and therefore the rapid growth of the IRL. The $E_{Mig}$ values were determined by CI-NEB method[33] and shown in Fig. 1c. The $E_{Mig}$ in $CoSb_2$ is similar to those in $XSb_y$, shown in Supplementary Fig. 1, indicating the Sb migration in $CoSb_2$ is not decisive. This is consistent with the experiment observation. According to the calculated results of $E_{Mig}$, we can preliminarily estimate the growth rate of the IRL. For examples, $ZrSb_2$ and $HfSb_2$ may grow faster, while YSb and $TaSb_2$ tend to grow slowly. The relative magnitude of $E_{Mig}$ is related to the strength of X–Sb chemical bonds, which can be reflected by the charge density between Sb atoms[34]. The calculated line profile of charge density from one Sb to the other Sb atom is shown in Supplementary Fig. 2. The sequence of charge density from high to low follows that of $E_{Mig}$. High charge density (i.e., tight bonding) between Sb atoms hinders the migration of Sb atom, while less density charge benefits the movement of Sb atom.

$E_{IR}$ and $E_{Mig}$, corresponding to interface reaction and diffusion, respectively, can be combined to serve as the criterion to screen promising barrier layer for $CoSb_3$-based SKD joint. The area above zero $E_{IR}$ corresponds to the condition of no bonding with

SKD (gray area). As a successful barrier layer material, the $E_{IR}$ should be negative to form the IRL. Otherwise, no chemical reaction and bonds occur, as well as the mechanical force between the SKDs and electrodes. In addition, the $E_{Mig}$ should be large enough to limit the Sb diffusion. Otherwise, the IRL grows too fast to increase contact resistance and perhaps destroy SKD materials. However, it is very hard to define a specific value for the $E_{Mig}$ as the critical point. Previous study shows that Ti is reasonably good barrier layer materials. Therefore, in this study, we assume that the $E_{Mig}$ should be similar to or larger than that for Ti. Combining these two effects, here we propose a 'sweet spot' area located in the right corner as marked by yellow color in Fig. 1d.

**Experimental verification and kinetic analysis**. Based on the above analyses, we select Nb and Zr to test in experiment. Nb is believed the promising candidate of barrier layer. The Nb joint and SKD/Zr joint (Zr joint) were fabricated and the interfacial structure evolution under accelerated aging conditions was systematically investigated. According to the elemental mapping images and compositional profile (Fig. 2b, Supplementary Fig. 3), the as-prepared interfaces of both Nb joint and Zr joint are composed of dense IRLs ($NbSb_2$ or $ZrSb_2$, respectively), and decomposed layer (DL) of $CoSb_2$, consistent with calculation results. Both IRLs grow without defects, while the growth of DL is accompanied by detectable defects (Supplementary Fig. 4). Trace of Yb elements can been found only at the DL/IRL interface, indicating Yb is not involved in the interfacial reaction and diffusion. However, due to the low content of Yb element (see Fig. 2a), it is hard to identify how many Yb elements exists in interface.

As mentioned above, the growth of the IRL can be explained by the reaction diffusion theory[29], which includes two consecutive steps, the chemical reaction and the element diffusion (Fig. 1a).

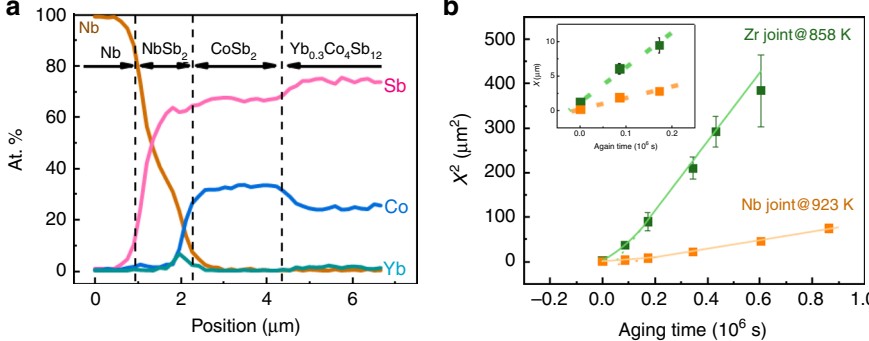

**Fig. 2 Interfacial microstructure characterization for Nb joint and Zr joint. a** Composition profiles of as-prepared Nb joint. At.% represents the percentage of each kind atom. **b** Correlation between aging time and square of IRL thickness ($x^2$) for Nb joint and Zr joint after aging at 923 K and 858 K, respectively. The insertion in **b** shows the linear relations between the IRL thickness and in initial aging time for Nb joint and Zr joint, indicating a chemical reaction control. Error bars represent the standard deviations of the IRL thicknesses.

**Table 1 The fitting kinetic parameters, $k_0$, $k_1$, $x_{1/2}$, $t_{1/2}$, $E_0$, and $E_1$ for Nb joint and Zr joint.**

| TE joint | Temperature (K) | $k_0$ ($10^{-12}$ m s$^{-1}$) | $k_1$ ($10^{-18}$ m$^2$ s$^{-1}$) | $x_{1/2}$ (μm) | $t_{1/2}$ (Hour) | $R^2$ | $E_0$ (kJ mol$^{-1}$) | $E_1$ (kJ mol$^{-1}$) |
|---|---|---|---|---|---|---|---|---|
| Nb | 823 | 1.0 | 0.4 | 0.4 | 188 | 0.99 | 202 | 311 |
|  | 848 | 1.7 | 2.2 | 1.3 | 447 | 0.82 |  |  |
|  | 873 | 5.6 | 10.4 | 1.9 | 183 | 0.87 |  |  |
|  | 898 | 7.5 | 28.9 | 3.9 | 287 | 0.95 |  |  |
|  | 923 | 25.3 | 67.9 | 2.7 | 59 | 0.96 |  |  |
|  | 948 | 43.3 | 153 | 3.5 | 45 | 0.99 |  |  |
| Zr | 823 | 12.1 | 83.3 | 6.9 | 319 | 0.91 | 225 | 242 |
|  | 838 | 24.8 | 194 | 7.8 | 176 | 0.83 |  |  |
|  | 848 | 49.2 | 393 | 8.0 | 90 | 0.90 |  |  |
|  | 858 | 79.7 | 658 | 8.3 | 58 | 0.99 |  |  |
|  | 873 | 102.7 | 857 | 8.4 | 45 | 0.95 |  |  |
|  | 898 | 187.7 | 1638 | 8.7 | 26 | 0.98 |  |  |

The growth kinetics of the IRL can be described by Deal and Grove's relation[35] (see Supplementary Note 1) giving as:

$$t = \frac{x - x_0}{k_0} + \frac{x^2 - x_0^2}{2k_1}, \quad (1)$$

where $t$ is time, $x$ is the thickness of the IRL, $x_0$ is the initial thickness of the IRL, $k_0$ is the chemical constant, and $k_1$ is the diffusional constant. $k_0$ and $k_1$ obey the Arrhenius relation given as: $k_0 = k_0^0 \exp(-\frac{E_0}{RT})$ and $k_1 = k_1^0 \exp(-\frac{E_1}{RT})$, respectively, where $k_0^0$ and $k_1^0$ are pre-exponential factor of chemical constant and diffusional constant, respectively, $E_0$ and $E_1$ are activation energy of chemical reaction and element diffusion, respectively, $R$ is molar gas constant. When the IRL is thin, the growth of the IRL is controlled by chemical reaction described by the linear item in Eq. (1). The chemical reaction (Fig. 1a) could be simply considered as: $2CoSb_3 + Nb(Zr) = 2CoSb_2 + NbSb_2(ZrSb_2)$. With the reaction proceeding the dominating mechanism gradually changes from chemical reaction control to element diffusion control.

During the mechanism switching, there is a critical thickness of the IRL, $x_{1/2}$, at which the diffusion time of Sb atoms is equal to the following chemical reaction time. Based on Eq. (1), $x_{1/2}$ is defined as: $x_{1/2} = \frac{k_1}{k_0} = \frac{k_1^0}{k_0^0} \exp\left(\frac{E_0 - E_1}{RT}\right)$, and the critical time, $t_{1/2}$, is the time when $x_{1/2}$ is achieved. Obviously, the IRL growth is controlled by chemical reaction when $x < x_{1/2}$ and by element diffusion when $x > x_{1/2}$. The reaction diffusion two-step mechanism is experimentally confirmed by the time dependence of squared thicknesses of IRLs of Nb joint and Zr joint (Fig. 2b). After aging for ~50 h, the time dependences of Nb joint aging at 923 K and Zr joint aging at 858 K demonstrate the relations of

$t \sim x^2$, indicating the diffusion-controlled mechanism. Nevertheless, at the beginning of aging, the relations of both Nb joint and Zr joint are $t \sim x$ (shown in the inset of Fig. 2b), indicating the reaction-controlled mechanism.

Based on a series of experimental thicknesses of IRL and kinetic equations above, the values of $k_0$, $k_1$, $x_{1/2}$, $t_{1/2}$, $E_0$, and $E_1$ for Nb joint and Zr joint at different aging temperatures are obtained (Supplementary Fig. 5), and listed in Table 1. The $k_0^0$ and $k_1^0$ of Nb joint are 4.52 m s$^{-1}$ and 32.58 m$^2$ s$^{-1}$, respectively, and the $k_0^0$ and $k_1^0$ of Zr joint are 2833 m s$^{-1}$ and 0.25 m$^2$ s$^{-1}$, respectively. The $k_0$ and $k_1$ represent the proceeding rate of chemical reaction and Sb diffusion, respectively, while the $E_0$ and $E_1$ represent the energy barrier (difficulty) of interfacial chemical reaction[36] and Sb migration. Comparing the obtained values of $k_0$, $k_1$, $E_0$, and $E_1$ for Nb joint and Zr joint, the chemical reaction and Sb diffusion in Zr joint proceed much faster than in Nb joint. These experimental results are also consistent with the trend of calculated activation energy barrier ($E_{Mig}$). However, it should be noted that the $E_{Mig}$ is the activation energy in the IRL, and the $E_1$ is the activation energy including all processes. Nevertheless, this result shows that the $E_{Mig}$ plays the important and perhaps dominant role, which is consistent with the assumption shown above.

By combining the Eq. (1) and Arrhenius relation, the full-time and full-temperature prediction surface of the IRL growth is achieved (Fig. 3) by fitting the experiment data. Due to the low $k_0$ and $k_1$, the predicted thickness of the IRL in Nb joint is much lower than that in Zr joint. The intersecting line between $x_{1/2}$ surface and prediction surface, as the boundary of the critical thickness and time, is varied with aging temperatures. Nevertheless, $t_{1/2}$ is usually <500 h for both joints aging at various

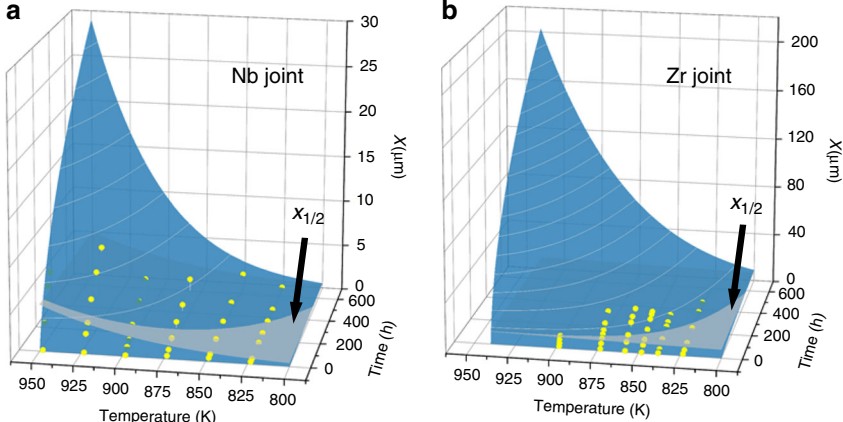

**Fig. 3 Prediction of IRL thicknesses.** Temperature and time-dependent IRL thickness ($x$) of Nb joint (**a**) and Zr joint (**b**). Yellow point: experimental data; blue surface: fitting of experimental data; and gray surface: critical thickness ($x_{1/2}$). The IRL growth is dominated by chemical reaction when $x < x_{1/2}$, and by element diffusion when $x > x_{1/2}$.

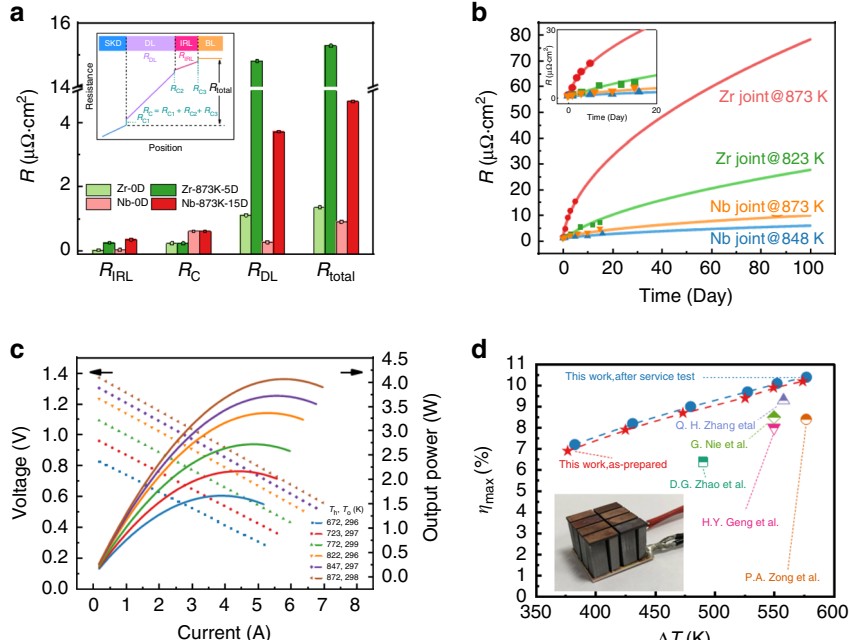

**Fig. 4 The interfacial resistivity and module performance. a** The interfacial resistivity ($R$) of Nb joint and Zr joint before and after aging at 873 K. Inset shows the diagram of interfacial resistivity analysis model (BL represents the barrier layer). $R_{IRL}$ and $R_{DL}$ are the contributions from IRL and DL, respectively. $R_C$ represents the sum of contact resistivities, $R_{C1}$ (SKD-DL), $R_{C2}$ (DL-IRL), and $R_{C3}$ (IRL-BL). **b** The fitting curves of the interfacial resistivity ($R$) of Nb joint aging at 848 K and 873 K, and the fitting curves of Zr joint, as comparison. **c** Open-circuit voltage ($V_{oc}$) and maximum output power ($P_{max}$) as a function of current at different operating temperatures of the as-prepared eight-pair module. **d** Maximum conversion efficiency ($\eta_{max}$) of the eight-pair module before and after long-term service test. Literature data of SKD single-stage module are also indicated[14,38-41].

temperatures. Considering the long-term service is usually above years, the thickness of IRL will reach $x_{1/2}$ at rather the early stage, that is to say, the element diffusion plays a dominant role eventually in whole service lifetime.

**Analysis and prediction of interfacial resistivity.** The interfacial resistivity ($R$) is essentially determined by the interfacial layers and contact in TE joints (Fig. 4a inset). The total interfacial resistivity ($R_{total}$) can be divided into three parts, $R_{total} = R_{IRL} + R_{DL} + R_C$ (Fig. 4a), where $R_{IRL}$ and $R_{DL}$ are the contributions from the IRL and the DL, respectively, which can be calculated by the layer thickness and the electrical resistivity like a series of

resistors. $R_C$ represents the sum of contact resistivities ($R_{C1}$, $R_{C2}$, and $R_{C3}$) between these layers that are majorly induced by the defects or the variations of Fermi-levels in these phases[37]. Therefore, $R_C$ can be regarded as a constant for given interfacial structures, which can be figured out by the intercept when linearly fitting the $R$ vs. $x$ curve, since $R_{IRL}$ and $R_{DL}$ are linearly increased when increasing $x$. In both Nb joint and Zr joint, the fitting results of $R_C$ are tiny (<0.6 $\mu\Omega$ cm$^2$). This is because the DL (CoSb$_2$) and the IRL (NbSb$_2$ or ZrSb$_2$) are zero-gap metal-like phases, and SKD is a heavily doped semiconductor. Unexpectedly but interestingly, it is found that bulk resistivities of the DL, $\rho_{DL}$, in the two joints are almost the same (~2900 $\mu\Omega$ cm) and much

larger than values of $\rho_{IRL}$. The large values of $\rho_{DL}$ probably result from the pores (see Supplementary Fig. 4) formed during the decomposition of $CoSb_3$ into $CoSb_2$. The experimentally obtained values of $R_C$, $R_{IRL}$, $\rho_{IRL}$, $R_{DL}$, and $\rho_{DL}$ of Nb joint and Zr joint aging at different temperatures are listed in Supplementary Table 2. The detailed analyses can be found in Supplementary Note 2 and Supplementary Figs. 6–8.

Figure 4a shows detailed contributions of total interfacial resistivity of Nb joint and Zr joint before and after aging. The $R_{total}$ of the as-prepared joints is mainly contributed by $R_C$ in principle, taking considerably low values (~1 $\mu\Omega\,cm^2$). After aging, $R_{IRL}$ and $R_{DL}$ increase owing to the growth of both the IRL and decomposition layer. However, $R_{DL}$ increases more rapidly than $R_{IRL}$, because of the formation of pores and cracks (see Supplementary Fig. 4) in the layer of $CoSb_2$. According to experimental results, $R_{DL}$ becomes the major contributor to $R_{total}$ for both Nb joint and Zr joint, as increasing aging temperature and aging period. In Nb joint, after 15 days aging at 873 K, $R_{DL}$ accounts for 80% $R_{total}$, and in Zr joint, after 5 days aging at 873 K, $R_{DL}$ accounts for 97% $R_{total}$.

By building the bridge between the microscopic kinetic parameters of the IRL growth and macroscopic resistance of interfacial components (IRL and DL), the dependency of the interfacial resistivity, aging temperature, and period, can be fitted and some typical fitting curves are shown in Fig. 4b. The fitted values agree well with the experimental data. The interfacial resistivity of Nb joint changes much more slowly than Zr joint, due to the lower diffusion rates and smaller $E_{Mig}$ of Sb in Nb joint. After 100 days aging at 873 K, the fitted interfacial resistivity of Nb joint is still <10 $\mu\Omega\,cm^2$, which will lead to a 2% rising of the total internal resistance. Meanwhile, the fitted interfacial resistivity reaches 80 $\mu\Omega\,cm^2$ in Zr joint after 100 days aging at 873 K, which will introduce a 16% increase of internal resistance. These results indicate that Nb joint would have much better performance during long-term service than Zr joint and even than previously studied SKD/Ti joints[28].

**Module performance and reliability**. We employed Nb as the barrier layer and fabricated an eight-pair module with optimized dimensions of 20 mm × 20 mm × 14.5 mm, using the $n$-type $Yb_{0.3}Co_4Sb_{12}$ and $p$-type $Ce_{0.85}Fe_3CoSb_{12}$ (Fig. 4d inset). TE properties of $p$-type and $n$-type materials are shown in Supplementary Fig. 9. According to full-parameter optimization simulation[17], the cross sectional areas of $p$-legs and $n$-legs are designed as 5 mm × 4 mm and 4 mm × 3 mm, respectively. Measurements of power output, internal resistance, and energy conversion efficiency were made in a home-made testing system. At the hot-side electrode temperature ($T_h$) of 872 K (temperature difference, $\Delta T$ of 574 K), maximum conversion efficiency ($\eta_{max}$), and maximum output power ($P_{max}$) of the module reach up to 10.2% and 4.1 W, respectively, which are record high values among the $CoSb_3$-based single-stage TE modules[14,38–41]. The long-term stability of the module was examined by 846-h test under the service condition of hot-side temperature at 818 K and cold-side temperature at 308 K. Changes of the output power ($P$) and the internal resistance ($R_{in}$) of the module are both <1%, indicating an excellent stability (Supplementary Figs. 10–12). The coefficient of thermal expansion (CTE) of Nb ($7.6 \times 10^{-6}\,K^{-1}$) and the $NbSb_2$ ($8.4 \times 10^{-6}\,K^{-1}$)[42] are relatively close to that of SKD materials (~$9.1 \times 10^{-6}\,K^{-1}$), which is one of the reasons for the excellent stability. Moreover, the thermal shock results (see Supplementary Fig. 13) and current kick results (see Supplementary Fig. 14) of Nb joint and the module also verify the stability. After the long-term service test, the maximum conversion efficiency and output power of the module still maintain 10.4% and 4.1 W, respectively,

at $T_h$ of 873 K ($\Delta T$ of 575 K; Fig. 4d). It is noteworthy that a large conversion efficiency of 9.5% can be obtained under $T_h = 823$ K, under which the module would exhibit an excellent service behavior with low degradation for long-term operation from the view of interfacial structure evolution as shown above.

## Discussion

The reaction diffusion model is successfully used to describe the evolution of $CoSb_3$-based-filled SKD/metal interfacial structure. Combining first-principles calculations and experimental investigation, it is revealed that, the formation and evolution of the IRL that determine the bonding behavior and interfacial resistivity are majorly contributed by the two processes: interfacial reaction between barrier layer and filled SKD, and Sb diffusion from filled SKD TE material across IRL. Regarding to the long-term service, the Sb diffusion plays the dominating role. Therefore, we proposed a criterion, the combination of interfacial reaction energy ($E_{IR}$) and Sb migration activation energy barrier ($E_{Mig}$), to predict both the bonding behavior and interfacial reliability of SKD/metal bonding. And then, we proposed a "sweet spot" in the $E_{IR} - E_{Mig}$ configuration to screen potential barrier layer materials with typical elements of Nb and Zr. The experimental results verify that Nb joint has the slowest IRL growth rate and smallest interfacial resistivity among all reported SKD/metal joints. An eight-pair $CoSb_3$-based SKD module using Nb as the barrier layer, fabricated in optimized dimensions, demonstrates a recorded conversion efficiency of 10.2% as single-stage TE modules at hot-side temperature of 872 K. Besides, the module shows an excellent stability by negligible degradations (<1%) of power output and conversion efficiency after 846-h long-term service test. This study not only offers an advanced $CoSb_3$-based SKD module with recorded conversion efficiency and excellent service stability, but also provides a simple criterion as an effective guidance on electrode interface optimization with bonding-blocking-conducting synergetic functions for TE device integration.

## Methods

**The fabrication and measurement of joints and module**. Nb joint and Zr joint were fabricated by one-step hot pressing (HP) sintering method. $Yb_{0.3}Co_4Sb_{12}$ ($n$-type SKD) and $Ce_{0.85}Fe_3CoSb_{12}$ ($p$-type SKD) TE materials was prepared by melting-quenching-annealing method. High purity raw materials were weighed in stoichiometric ratios and sealed in evacuated quartz tubes. The tubes were heated to 1353 K and maintained at this temperature for 24 h, then the tubes were quenched by water bath and annealed at 973 K for 120 h. Finally, the ingot was ground into fine powders.

For TE joint, SKD powder, Nb foil (Alfa Aesar, 99.8%, 25 ± 15 μm) or Zr foil (China New Metal Materials Technology, 99.9%, 25 μm) were loaded into a graphite die of 30 mm in inner diameter sequentially, and sintered at 923 K under 60 MPa for 30 min. The thicknesses of Nb foil and Zr foil were selected to be ~25 μm. The as-prepared joints were cut into dices in dimensions of 4 mm × 4 mm × 4.5 mm by the electric-spark machine. All dices were sealed into quartz tubes in vacuum and placed in furnaces at different annealing temperatures for the accelerating experiment. There are at least three parallel specimens in one tube.

The interfacial microstructures and chemical compositions of joints were investigated using a scanning electron microscope (SEM, ZEISS Supra 55) and its energy-dispersive spectrometer attachment. The thicknesses of layers were determined with the assist of measurement tools in SEM images and were presented after averaged from over ten randomly selected locations of each parallel specimen. The interfacial resistivity was measured at a home-built four-probe platform, and was presented after averaged from at least three locations of each parallel specimen. The principle of measurement was demonstrated in previous work[30].

The HPed $p$-type TE joints and $n$-type TE joints were cut into 5 mm × 4 mm and 4 mm × 3 mm, respectively. The hot side of $n$–$p$ couples was connected to $Mo_{50}Cu_{50}$, using Ag–Cu–Zn welding alloy, and the cold side was connected by copper metallized ceramic substrate, using Pb–Sb welding alloy. The module consisted of eight $n$–$p$ couples, 16 legs. Glass fibers were filled between legs. The measurements of the module were conducted in a home-made testing system described in previous work[17].

**Calculation details**. All calculations were performed at density functional level of theory, using the Vienna Ab initio Simulation Package[43]. A plane-wave basis and projector augmented-wave method pseudopotentials were used[44]. The Perdew–Burke–Ernzerhof generalized gradient approximation was adopted to treat exchange-correlation effects[45]. A cutoff of 370 eV was imposed on the kinetic energy. Sampling of the Brillouin zone was performed using gamma-only or $2 \times 2 \times 2$ Monkhorst-Pack grids according to the supercell size[46]. The accuracy on the total energy was set to $10^{-8}$ eV. Atomic and lattice parameter relaxations were optimized by a conjugate-gradient algorithm with an imposed numerical threshold of 0.005 eV Å$^{-1}$. $E_{IR}$ is defined as the formation enthalpy that are determined from chemical reaction $CoSb_3 + X \rightarrow CoSb_2 + XSb_y$. The $XSb_y$ is Sb-rich compound from the stable X–Sb compounds based on the Materials Project database[47]. And the activation energy for Sb atom migration ($E_{Mig}$) to neighboring vacancy are determined by CI-NEB method with eight inserted images[33].

## Data availability

All data generated are available from the corresponding author on reasonable request. The raw data contained in Figs. 1–4, Table 1, Supplementary Figs. 1–5 and 9–14, and Supplementary Tables 1 and 2 are available upon request from R.L. and S.B.

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

## Acknowledgements

This work was financially supported by the National Key Research and Development Program of China (grant no. 2018YFB0703604), National Nature Science Foundation of China (NSFC; grant nos. 51632010, 51972324, and 51971237), and Youth Innovation Promotion Association CAS (no. 2019253)

## Author contributions

R.L. and S.B. designed the research project and supervised the experiment. J.C., C.W., Q.Z., and X.X. conceived and fabricated TE joints, and module. J.C. and M.G. carried out investigations of interfacial microstructures. J.C. and J.L. characterized the module and TE joints. J.H. performed calculations. J.C., J.H., X.S., L.C., R.L., and S.B. wrote and edited the manuscript. All authors contributed to the data analysis, discussed the results, and commented on the manuscript.

## Competing interests

The authors declare no competing interests.
