## [Peer Review File · Nature Communications]

Reviewers' comments:

Reviewer #1 (Remarks to the Author):

The paper on the "Electrode interface optimization for CoSb₃-based Skutterudite thermoelectric devices.." is certainly an interesting contribution to a successful Skutterudite module fabrication. The paper is well documented, length and discussion are appropriate. There are a few items to be considered prior to publication:

- 1) The manuscript needs overhaul by a NATIVE ENGLISH SPEAKER (although it is well written in general, articles are missing and formulations need to be honed).
- 2) The authors state "It has been widely observed experimentally that filler atoms in CoSb₃-based filled SKD such as Yb, Ba et al. are not involved in the growth of the IRL". This is not true as we can simply track intermetallic compounds from databases such as: "Ba₅{V,Nb}12Sb_{19+x}, novel variants of the Ba₅Ti₁₂Sb_{19+x}-type: Crystal structure and physical properties" in Phys. Chem. Chem. Phys. (2015) 17, 24248-24261 etc...
- 3) Although thin layers are not prone to exert stress due to lattice parameter mismatch, no discussion was made on this topic although experimental data for the intermetallic layers NbSb₂, NbSb exist.
- 3) Fig 1b: Formation energies depicted are from the binary compound richest in Sb (such as NbSb₂..) or from the thermodynamically most stable binary antimonide??

Reviewer #2 (Remarks to the Author):

It is an interesting study to solve an overlooked area in thermoelectric system. However, tests have only been conducted for about 100 days, far shorter than real application scenario. Also test of constant thermal shock should also be conducted over long time. This type of study may fit basic materials science journal better.

Reviewer #3 (Remarks to the Author):

This manuscript proposed a criterion for screening barrier layer, used in thermoelectric devices, based on the combination of the moderately low interfacial reaction energy (EIR) and high migration energy barrier (EMig) of Sb through the formed interfacial reaction layer (IRL). It's an interesting research work on making thermoelectric devices and meaningful to advance the application of TE techniques. However, I think the manuscript needs major revision before consideration for publication, addressing problems such as the following:

1. Were the interface studies done simply under heating? How does the current/ voltage kick in during service affect the chemical reaction and diffusion process?
2. In page 8, statement of 'when the IRL is thin, the growth of the IRL is controlled by chemical reaction described by the linear item in Eq. (1)' seems to contradict with the following statement 'after aging for about 50 hours, the time dependences of Nb joint aging at 923 K and Zr joint aging at 858 K demonstrate linear relation, indicating the diffusion control mechanism'. Why ?
3. Double check the statement in line 210-212. The values of those constants for Zr case do not look reasonable.
4. Since the resistivity of defect layer dominates the resistivity at the interfaces, does that mean

inhibiting the consumption of Sb is actually the final goal here, which is determined by the steady state eventually achieved when the depletion of Sb is complete? In that case, how long will the steady state be achieved and is it long enough compared to the test duration here and to service duration required in the future?

5. How does the temperature affect the chemical reaction and diffusion process and how was it reflected in the parameters such as EIR, EMig, and Formation Enthalpy?

6. In figure S4, the original interfaces before aging test should be provided as references. Are there voids formed at SKD side near the interface?

Reply to the first Referee's report:

Question 1:

The manuscript need overhaul by a NATIVE ENGLISH SPEAKER (although it is well written in general, articles are missing and formulations need to be honed.

Response:

Thank you for your suggestion. We have revised articles and formulations in a view of native English speaker.

Question 2:

The authors state "It has been widely observed experimentally that filler atoms in CoSb₃-based filled SKD such as Yb, Ba et al. are not involved in the growth of the IRL". This is not true as we can simply track intermetallic compounds from databases such as: "Ba₅Ti₁₂Sb_{19+x}, novel variants of the Ba₅Ti₁₂Sb_{19+x}-type: Crystal structure and physical properties" in Phys. Chem. Chem. Phys. (2015) 17, 24248-24261 etc...

Response:

Sorry for the mistakes. We agree with the referee that the fillers may be involved in the growth of IRL such filler Ba. In this study, we used the Yb-filled skutterudites as an example. Due to the low filling fraction of Yb in CoSb₃ and the small diffusion coefficient of Yb, we did not obviously observe that the Yb is involved in the growth of the IRL. Therefore, the filler Yb is not considered in the current paper. Nevertheless, we have cited the paper and modified the statement in the revised manuscript.

Question 3:

Although thin layers are not prone to exert stress due to lattice parameter mismatch, no discussion was made on this topic although experimental data for the intermetallic layers NbSb₂, NbSb₂ exist.

Response:

Thanks for your helpful advice. The coefficient of thermal expansion (CTE) of the NbSb₂ ($8.4 \times 10^{-6} \text{ K}^{-1}$, *Intermetallics* 65, 94-110, 2015) is relatively close to that of SKD materials (around $9.1 \times 10^{-6} \text{ K}^{-1}$). Therefore, the Nb is very suitable as barrier layer material for skutterudites. We have added the CTE data of NbSb₂ and related discussions into the revised manuscript.

Question 4:

Fig 1b: Formation energies depicted are from the binary compound richest in Sb (such as NbSb2..) or from the thermodynamically most stable binary antimonide??

Response:

It is the Sb-richest phase. There are two reasons for this assumption. (1) In this study, as shown in the answer for question 2, Yb is not diffusive as compared with Sb. Therefore, it is the diffusion of Sb to dynamically determine the chemical reactions. (2) The thickness of IRL is extremely small as compared with the vast thickness of skutterudites. The above two reasons lead to the formation of Sb-richest phase, which has also well confirmed by experiment observation. We have updated and modified the statement in the revised manuscript.

Reply to the second Referee's report:

Question 1:

It is an interesting study to solve an overlooked area in thermoelectric system. However, tests have only been conducted for about 100 days, far shorter than real application scenario. Also test of constant thermal shock should also be conducted over long time. This type of study may fit basic materials science journal better.

Response:

Thanks for your interest and important suggestions. Since thermoelectric devices are expected to service over years or even tens of years, it is impossible to conduct the identical service life test in laboratory at service condition. Therefore, we actually carry out an accelerated test in current study to shorten the testing procedure time. Since the Sb atom can easily sublime at high temperature above 800 K (*J Alloy Compd* **509**, 3166-3171, 2011), to achieve long life in real application, the hot side temperature at which SKD devices are expected to work should not exceed 823 K. In current study, the temperature of 873 K we used for the estimation of interfacial resistivity is much higher than the real working conditions. According to our reaction-diffusion model, after 100 days aging at 873 K, the interfacial resistivity of Nb joint will reach the value of $10 \mu\Omega \text{ cm}^2$, which corresponds to the fitted data aging at 823 K for 3.2 years or aging at 773 K for 87 years, as shown in following **Fig. R1 a**.

Therefore, from this point of view, 100 days test at 873 K in lab is adequate to demonstrate the stability of the SKD device in real application scenario.

Besides of the previous isothermal aging, thermal shock test was supplemented. As shown in **Fig. R1 b**, during the thermal cycling test, the cold-side temperature (T_c) was fixed at 301 K, and the hot-side temperature (T_h) was shifted between 810 K and 483 K at the heating/cooling rate of 20 K/min. Similar with the previous results, the module with the Nb diffusion barrier shows a good reliability in thermal shock test. The maximum output power (P_{max}) and the internal resistance (R_{in}) of the tested module show less than 4.2% and 0.5% changes, respectively, after 300 cycles. The results of the thermal cycling test were added into the supplementary information in the revised version.

Of course, for the engineering application of TE devices, more systematical investigation on the service behavior should be conducted. The current experimental results showed that the Nb-bonded module exhibits excellent service behavior with very low degradation for long-term operation under T_h of 827 K from the basic view of interfacial structure evolution, under which a high conversion efficiency of 9.5% can be obtained. We have added these data and figures into the supplementary materials in the revised manuscript.

Fig. R1 a The predicted interfacial resistivity (R) of Nb joint at 873 K, 823 K and 773 K based on reaction-diffusion model. **b** The maximum output power (P_{max}) and the internal resistance (R_{in}) of the 8-pair module under constant-thermal-shock condition that hot-side temperature (T_h) cycles between 810 K and 483 K and cold-side temperature (T_c) is fixed at 301 K.

Reply to the third Referee's report:

Question 1:

Were the interface studies done simply under heating? How does the current/ voltage kick in during service affect the chemical reaction and diffusion process?

Response:

Thanks for your important concern. In current work, the thermodynamic study on the electrode joints was done only under heating. But the long-term test of TE module was carried out under the condition with current kick, which demonstrated the good stability of module. To clarify the influence of current on the interfacial behavior of SKD and the performance of TE module, we demonstrate the tests for SKD/Nb joint (Nb joint) and TE module with Nb barrier layer under large current stage. As shown in **Fig. R2 a**, after applying an increasing current density up to the current maximum output power achieved ($\sim 38 \text{ A}\cdot\text{cm}^{-2}$) for 5 minutes at each point with the step of every 10 minutes at 773 K, the internal resistivity of Nb joint obviously shows no change. Furthermore, the interfacial structures of the Nb joint show no change after applying current test (see **Fig. R2 b**). We further tested the TE module with the current closed to the shorten current at the hot-side temperature of 773 K. As shown in **Fig. R2 c**, the internal resistance of TE module under large current was tested about every 20 hours and there are obviously no changes during 150 hours testing. All these data reveals that the current/voltage has a limited effect on the chemical reaction and diffusion process of SKD-based interfaces.

Certainly, for the engineering application of TE devices, more systematical investigations on the service behavior, including the stability under current/voltage shock, mechanical shock and operation atmosphere as well, should be conducted. The current work really demonstrated the great possibility of its excellent service behavior through the basic view of interfacial structure evolution and performance of TE module. We have added these data into the supplementary materials in the revised manuscript.

Fig. R2 **a** Correlation between the current density (J) and relative electrical resistance variation (R/R_0) for Nb joint with length of 4 mm. **b** The interfacial microstructures for Nb joint before and after applying current test. **c** Time dependence of the internal resistance for SKD-based module with Nb barrier under the current of 5.2 A at the hot-side temperature of 773 K.

Question 2:

In page 8, statement of ‘when the IRL is thin, the growth of the IRL is controlled by chemical reaction described by the linear item in Eq. (1)’ seems to contradict with the following statement ‘after aging for about 50 hours, the time dependences of Nb joint aging at 923 K and Zr joint aging at 858 K demonstrate linear relation, indicating the diffusion control mechanism’. Why ?

Response:

Thanks for your question. Sorry for the misdescribing. In **Fig. 2 b**, the y-axis is labelled by the square of IRL thickness (x^2), instead of IRL thickness (x). According to the reaction-diffusion model (Eq. 1 in main text), the linear relation ($t \sim x$) indicates the chemical-reaction-controlled mechanism while quadratic relation ($t \sim x^2$) indicates the diffusion-controlled mechanism. We have revised the **Fig. 2b** including the insertion

and the figure caption to clarify the relation.

Question 3:

Double check the statement in line 210-212. The values of those constant for Zr case do not look reasonable.

Response:

Thanks for your suggestion. In fact, the interfacial reaction layer (IRL) in Zr joints grows much faster than in Nb joints at aging temperatures. As shown in **Supplementary Figure 4**, inconstant boundary of the IRL in Zr joints makes it difficult to precisely characterize the IRL thicknesses for Zr joints, especially when aging at higher temperature. In order to improve the data reliability for Zr joints, we supplement three new batches of Zr joints aging at 858 K, 873 K and 898 K, respectively. We measured the thickness of the IRL for these new specimens and updated all experiment data as shown in **Revised Table 1**. The revised k_0^0 and k_1^0 of Zr joint are 2833 m s^{-1} and $0.25 \text{ m}^2 \text{ s}^{-1}$, which are more reasonable than before. Importantly, the main conclusions of the discussion are not changed by the updated data. The related statements, tables and figures have been updated as well in the revised version.

Revised Table 1 The fitting kinetic parameters, k_0 , k_1 , $x_{1/2}$, $t_{1/2}$, E_0 , E_1 for Nb joint and Zr joint.

TE joint	Temperature (K)	k_0 (10^{-12} m/s)	k_1 ($10^{-18} \text{ m}^2/\text{s}$)	$x_{1/2}$ (μm)	$t_{1/2}$ (Hour)	R^2	E_0 (kJ/mol)	E_1 (kJ/mol)
Nb	823	1.0	0.4	0.4	188	0.99	202	311
	848	1.7	2.2	1.3	447	0.82		
	873	5.6	10.4	1.9	183	0.87		
	898	7.5	28.9	3.9	287	0.95		
	923	25.3	67.9	2.7	59	0.96		
	948	43.3	153	3.5	45	0.99		
Zr	823	12.1	83.3	6.9	319	0.91	225	242
	838	24.8	194	7.8	176	0.83		
	848	49.2	393	8.0	90	0.90		
	858	79.7	658	8.3	58	0.99		

873	102.7	857	8.4	45	0.95
898	187.7	1638	8.7	26	0.98

Question 4:

Since the resistivity of defect layer dominates the resistivity at the interfaces, does that mean inhibiting the consumption of Sb is actually the final goal here, which is determined by the steady state eventually achieved when the depletion of Sb is complete? In that case, how long will the steady state be achieved and is it long enough compared to the test duration here and to service duration required in the future?

Response:

Thanks for important concerns. Yes, we agree that the major goal is to employ barrier layer to obstruct the Sb diffusion. However, in the reaction-diffusion process, there is no theoretically steady state. In fact, this process will not stop unless the reactants, i.e. CoSb₃ matrix or barrier layer (e.g. Nb), are exhausted. In order to determine whether the amount of Nb barrier is enough to satisfy the service life, we calculate the consumption rate of Nb barrier at different temperatures based on kinetic analysis, and show the results in **Fig. R3**. The Nb barrier used in current study is about 25 μm. In order to consume all Nb barrier layer, it will take 13.2 years at 873 K and more than 100 years when the temperature is lower than 823 K. Since the service temperature for SKD-based module is usually designed to be lower than 823 K, the Nb barrier layer with 25 μm can meet the requirement of long-term service in real application.

Fig. R3 Predicted Nb-consumed thickness curves for Nb joint aging at 873 K, 823 K and 773 K.

Question 5:

How does the temperature affect the chemical reaction and diffusion process and how was it reflected in the parameters such as E_{IR} , E_{Mig} , and Formation Enthalpy?

Response:

Thanks for important concerns. Temperature does have effects on both the chemical reaction and diffusion process. As discussed in the manuscript, k_0 is the chemical constant and k_1 is the diffusional constant, and both of them are temperature dependent following the Arrhenius relation given by $k_0 = k_0^0 \exp(-\frac{E_0}{RT})$ and $k_1 = k_1^0 \exp(-\frac{E_1}{RT})$. High temperature definitely leads to quick chemical reaction and fast atomic diffusion. The E_{IR} and E_{Mig} are calculated at 0 K by *ab initio* method. In principle, E_{IR} and E_{Mig} reflect the energy barrier when reaction and diffusion need to overcome, which are determined by the chemical bond circumstance. With temperature increasing, the bond strength may slightly weaken caused by thermal vibration. As reviewed by Schlesinger (*Chem. Rev.* **113**, 8066–8092, 2013), the change of formation enthalpy with temperature is insignificant in antimonides. It is difficult for *ab initio* calculation to directly investigate the activation energy for the chemical

reaction. However, following Eyring's reaction rate theory (*Chem. Phys.* **3**, 107-115, 1935), the E_{Mig} is directly correlated to the activation energy for atomic diffusion and has a good agreement with our experimental parameters E_1 . It should be noted that the high temperature also leads to the large entropy, which may change the thermodynamic state. But this beyond the scope of current study and needs further study in the future.

Question 6:

In figure S4, the original interfaces before aging test should be provided as references. Are there voids formed at SKD side near the interface?

Response:

Thanks for important suggestion. We have added microstructures pictures of the original interface for Nb joint and Zr joint before aging test to **Supplementary Figure 4**. The as-sintered microstructures, shown in **Fig. R4**, demonstrate that there are no voids formed at SKD side for Nb joint and Zr joint.

Fig. R4 microstructures before aging for Nb joint Zr joint. There are no voids at the SKD side before aging for both joints.

Summary of changes:

1. The statement for the description of fillers in skutterudites has been revised in line 6-10 of the second paragraph on page 4.
2. The descriptions on why the binary products in the interfacial chemical reactions are Sb-richest phase are revised in line 19 on second paragraph on page 4 and line 1-4 on first paragraph on page 5.
3. The description for Fig. 2b in line 7-9 on second paragraph on page 8 is revised.
4. The x -axis title of Fig. 2b is renamed as aging time for better understanding.
5. The insert figure in Fig. 2b is revised to demonstrate the linear relations between the IRL thickness and aging time for Nb joint and Zr joint.
6. The y -axis of Fig. 3 is changed from 10^6 second to hour to match the statement in the text.
7. The statement on pre-exponential factor of chemical constant and diffusional constant is updated in line 3-5 on third paragraph on page 8.
8. The statement on critical thickness of the IRL, $x_{1/2}$, in line 1-2 on first paragraph on page 10, is deleted based on the revised experimental data.
9. Fig. 2b, Table 1, Fig. 3b, Fig. 4a and b, Supplementary Fig. 5 to 8 and Supplementary table 2 are updated.
10. The coefficient of thermal expansion (CTE) data of NbSb₂ and related discussions are added into line 6-8 on first paragraph on page 13.
11. The microstructure images for Nb joint and Zr joint before aging are added to Supplementary Fig. 4 and the corresponded figure caption is revised.
12. Supplementary Fig. 13 has been added to show the constant thermal shock results for TE module.
13. Supplementary Fig. 14 has been added to show the stability of Nb joint and the module with Nb barrier layer under the condition of the current kick.
14. The descriptions on constant thermal shock results and current kick results are added in line 8-10 on first paragraph on page 13.
15. The y -axis title of Supplementary Fig. 1 is renamed as activation energy.
16. Only lines about Sb-Sb channel are kept in Supplementary Fig. 1 and the corresponded figure caption is revised.
17. The sign for Zr element is added to the Zr joint mapping in Supplementary Fig. 3b.
18. The unit of temperature in Supplementary Fig. 10 is changed from Degree Celsius to Degree Kelvin.

19. The definition of “R” in Arrhenius relation has been added as “R is molar gas constant” in line 8 on first paragraph on page 8.
20. Experimental details are revised for better understanding.
21. The description on cause of contact resistivities (R_c) in line 5-7 on second paragraph on page 10 is revised as the defects or the variations of Fermi-levels in these phases in order to be more precisely.
22. The definition of interfacial reaction energy (E_{IR}) is updated and the related discussions and figures (Fig. 1) are modified accordingly. Although the difference between the old and new E_{IR} is quite small, the revised E_{IR} is more suitable for the case in this study and thus well consistent with the experiment results. The negative E_{IR} is the criterion for the occurrence of chemical reaction between CoSb_3 and metal X. Nevertheless, the conclusions are not affected by such modification. The description and discussion on interfacial reaction energy have been revised in line 16-19 on second paragraph on page 4 and line 4-7 on first paragraph on page 5. Original Fig. 1b and the descriptions of “open element methodology” are deleted.
23. The brief description on the screening criterion for potential barrier layer candidates is added into Fig. 1 caption.
24. Calculation details are revised according to the new definition of interfacial reaction energy.
25. The original migration energy barrier (E_{Mig}) is renamed as activation energy barrier of Sb migration (E_{Mig}) to prevent misunderstanding.
26. The schematic diagram of the initial chemical reaction is added into Fig. 1a to fully demonstrate the interfacial reaction layer growth.
27. The visual effect of schematic diagrams in Fig. 1a has been update vividly.
28. The value of interfacial reaction energy (E_{IR}) has been replaced by that of the original formation energy Supplementary Table 1.
29. All changed details, mainly including language polish but not mentioned above, are marked with red in the revised manuscript.

We thank the three reviewers again for their careful considerations and very helpful suggestions. We believe that we have made clear explanations and proper modifications, and that the paper should now be suitable for publication in *Nature Communications*.

Sincerely yours,

Dr. Ruiheng Liu and Prof. Shengqiang Bai

Shanghai Institute of Ceramics, Chinese Academy of Science

1295 Dingxi Road, Shanghai, 200050 China

Tel: 86-21-69163686

Email: liurh@mail.sic.ac.cn (R. Liu), bsq@mail.sic.ac.cn (S. Bai)

REVIEWERS' COMMENTS:

Reviewer #2 (Remarks to the Author):

Good for being accepted for publication now.

Reviewer #3 (Remarks to the Author):

The authors provided sufficient justifications in the revised manuscript to address most of my initial comments on the first version of the manuscript.

I have no further comments at this point and the manuscript should be ready for publication

Reply to the second Referee's report:

Comment:

Good for being accepted for publication now.

Response:

We appreciate for your previous helpful suggestions and comments.

Reply to the third Referee's report:

Comment:

The authors provided sufficient justifications in the revised manuscript to address most of my initial comments on the first version of the manuscript.

I have no further comments at this point and the manuscript should be ready for publication

Response:

The manuscript benefits from your previous interest and important suggestions.
Thank you.

We thank the three reviewers again for their careful considerations and very helpful suggestions. We believe that we have made clear explanations and proper modifications, and that the paper should now be suitable for publication in *Nature Communications*.